# Imaging Diagnostics of Inside of a Building Wall Using Millimeter-Wave Reflectometer

**Shota Osaki** [1,*]**, Atsushi Mase** [2] **, Yoshikazu Hirata** [3] **and Munehiro Iwakura** [1]

1    Kyushu Keisokki Co., Ltd., Fukuoka 8120015, Japan; mune@qk-net.co.jp
2    Global Innovation Center, Kyushu University, Kasuga 8168580, Japan; mase@gic.kyushu-u.ac.jp
3    Daisue Construction Co., Ltd., Tokyo 1368517, Japan; y-hirata@daisue.co.jp
*    Correspondence: ohsaki@qk-net.co.jp

**Featured Application: Non-destructive exterior wall diagnostic device.**

**Abstract:** Progress in microwave and millimeter-wave technologies has enabled advanced diagnostics for industrial applications. The transmission, reflection, scattering and radiation processes of electromagnetic waves are utilized as diagnostic principles. Specifically, the reflectometric method has gained importance in various applications due to the possibility of the high localization and accessibility of measurements, as well as the non-destructive nature of the systems. In this paper, radar reflectometers were applied to the measurement of the inside of a building wall, that is, the inspection of tile materials attached to a concrete wall. The measurement principle utilizes the phase interference effect of the reflected wave due to the multiple reflections between the two layers (Fabry–Perot effect). The results show the imaging inside the surface related to the peering condition between the tile and concrete wall, and the quantitative evaluation of the condition with non-destructive inspection.

**Keywords:** millimeter-wave; reflectometer; non-destructive inspection; imaging; industrial application

## 1. Introduction

In Japan, tile materials are attached to make the outer wall of a building look attractive, and to protect the concrete wall. The detachment of tile materials may occur due to adhesion defects and aging, which may cause dangerous accidents. Therefore, the inspection of the adhesive condition, as well as the peeling state, is of importance. Conventionally, a hammering sound inspection is utilized, which detects invisible defects under the attached tile by striking the surface with a hammer and listening to the sound. An inspector judges the state by the sound. Usually, veteran engineers are engaged in this work. This method does not need a sophisticated system and is a non-destructive measurement. However, it has serious problems: for example, it requires skilled experts and is time-consuming. There is no convenient way to quantitatively evaluate the results. As a further note, organic adhesive exterior wall tiles have become more widespread in recent years. It is difficult to detect the adhesion state of these tiles by using a hammering inspection.

Recently, studies of hammering robots have been progressing [1–3]. The robot imitates the hammering sounds of skilled inspectors. The hammering sound is evaluated by inspectors remotely or analyzed by using a computer. Hammering robots belong to the remote measurement systems, since one can control the system and analyze the resulting sound data remotely. However, the above-mentioned difficulties with the manual hammering inspection still exist. Additionally, considering dangerous activities, such as building walls and other constructions, the application of non-contact, as well as non-destructive, diagnostic systems should be expected.

As an alternative to the hammering method, electromagnetic wave diagnostics using infrared waves has been studied [4–10]. Infrared thermography has advantages, since it is a perfectively non-intrusive and non-contact measurement method. However, it still has an

issue in the quantitative diagnosis of the tile adhesion measurement since it can be difficult to understand the results due to ambient thermal effects, as well as the lack of spatial resolution. Although not commonly used for exterior wall diagnosis, ultrasonic methods are also available as a non-destructive testing technique [11,12]. The ultrasonic C-scan obtains the depth-directed damage distribution. However, it is difficult to apply to exterior wall diagnostics because the sample must be immersed in liquid. Air-coupled ultrasonic has also been studied, but there is still no standardized method for detecting internal defects in exterior walls with pulse echoes. Microwave/millimeter-wave reflectometric measurement has also gained importance in the present measurement due to its non-invasive nature. Two characteristics of microwave/millimeter-wave diagnostic systems combine to act effectively. The first is the ability to penetrate dielectric materials. The second is the use of sensitive phase fluctuation measurements [13]. In this work, a type of radar reflectometer was first applied for the measurement, which utilizes the phase interference effect of reflected waves due to the multiple reflection between the two layers (Fabry–Perot effect). The results show the imaging inside the surface related to the peering condition between the tile and concrete wall, and the quantitative evaluation of the condition.

In Section 2, the diagnostic principle using Fabry–Perot interferometry is briefly described. An initial measurement is compared with a theoretical calculation. The practical hardware system together with the signal processing of the results is described in Section 3. In Section 4, the experimental results using various simulated samples and quantitative evaluation of the tile adhesion condition are provided, followed by the conclusion.

## 2. Reflectometric Measurement

### 2.1. Principle

A Fabry–Perot interferometer consists of two parallel reflecting surfaces, as shown in Figure 1a. When an electromagnetic wave is incident on the surface, the wave is partially transmitted and reflected due to the differences in the refractive index. The transmitted and reflected waves depend on the interference between the multiple reflections at the two reflecting surfaces. When the transmitted beams, $T_1$, $T_2$, ----, are in phase, the sum of the sequences generates a high transmission peak, and the sum of the reflected waves, $R_1$, $R_2$, ----, shows the minimum transmission peak. On the other hand, when the transmitted beams are out of phase, the sum of them shows the minimum value, and the reflected waves show the maximum. In Figure 1a, this interference depends on the wavelength of the electromagnetic wave $\lambda$, the incident angle to the surface $\theta$, the thickness of the dielectric material between the reflecting surfaces $l$ and the refractive index of the material $N = \sqrt{\varepsilon}$ ($\varepsilon$ is the relative permittivity). For the inspection of tile materials attached to the outer wall of the building, we can utilize the principle of Fabry–Perot interferometry. The model of a subject is shown in Figure 1b. The reflectance function is as follows:

$$I_r \propto \frac{2R(1 - \cos \delta)}{1 - 2R \cos \delta + R^2} \tag{1}$$

where $R$ is the reflectivity of the front and back surfaces of the tile as follows:

$$R \cong \left( \frac{1 - N}{1 + N} \right)^2. \tag{2}$$

Additionally, $\delta$ is the phase difference between successive transmitted waves $T_1$, $T_2$, ----, which is as follows:

$$\delta = \left( \frac{2\pi}{\lambda} \right) 2Nl \cos \theta \tag{3}$$

Therefore, the reflected wave is minimized when the value of $2Nl\cos\theta$ is equal to an integral multiple of the wavelength ($\lambda$, $2\lambda$, ----). On the other hand, the reflected wave is maximal when it is equal to an odd multiple of half a wavelength ($\lambda/2$, $3\lambda/2$, ----). Since

the thickness of the tile plate is given, the maximum and minimum changes in the reflected wave can be obtained by changing the wavelength of the incident wave.

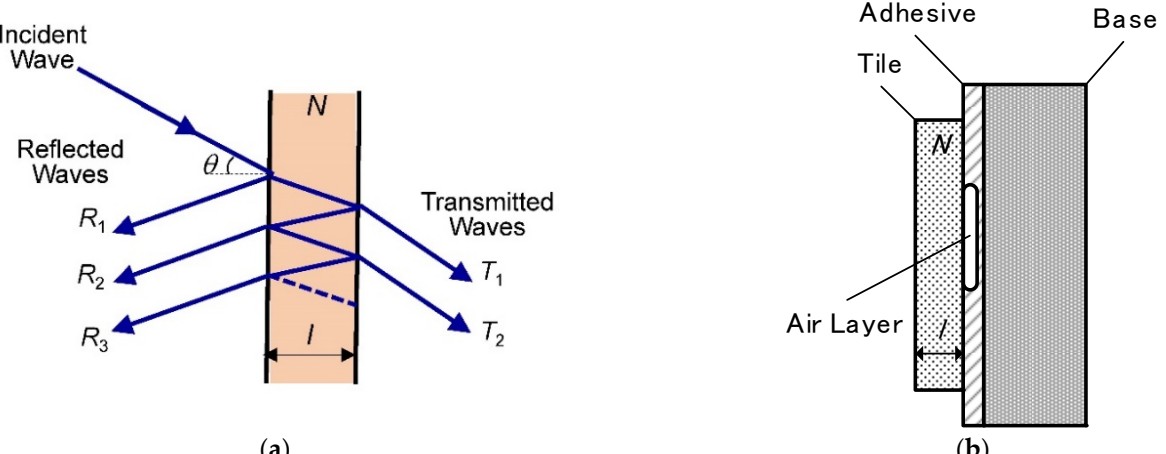

(**a**)                                                                                           (**b**)

**Figure 1.** (**a**) Model of Fabry–Perot interferometry; (**b**) model of present subject.

### 2.2. Principle Verification

The initial experiment was performed in order to identify the existence of the above interferometry effect. The transmitting and receiving antennas were installed on a tile surface (Figure 1b) and connected to port 1 and port 2 of a vector network analyzer (VNA), respectively. The reflected wave from the tile surface was evaluated using the $S_{21}$ parameter of the VNA. Probe-type waveguide antennas (WR-19:40-60 GHz) and/or substrate antennas (TSA-23T58G-01 (product of EM-WISE)) were used as a transmitter and a receiver. The antennas' angle was 30 degrees, and the distance between the antenna tip and the tile was approximately 5 mm. Considering the effective lengths of the above antennas, this distance is located near the far-field condition. In Figure 2, the measurements are plotted as a function of the millimeter-wave frequency. The solid line in Figure 2 is an example of a numerical calculation of Equation (2). Min-max normalization was performed to make the graph comparison easier. Here, the parameters of the tile thickness $l$ = 7.3 mm, the incident angle $\theta$ = 10° and the refractive index $N$ = 2.19 were assigned. A good agreement between the experimental results and the numerical calculation was obtained. In a real subject, the interferometry effect is not as simple as explained in the concept of Figure 1a, since the subject has a multilayer structure, as shown in Figure 1b. The dielectric constant (relative permittivity) of each layer is said to be as follows: tile 5–6, organic adhesive 4–6, concrete 6–8. Therefore, the reflection between the tile and the air is much bigger than the reflection between the tile and the organic adhesive, and between the organic adhesive and the concrete. The measured reflection model is rather close to two-layer reflection between the tile and the air. The result of Figure 2 shows that the two-layer reflection effect can be utilized in the present study.

### 3. Practical Application

### 3.1. System Fabrication

A prototype of the measurement system was fabricated to prepare it for practical use. A schematic diagram of the system is shown in Figure 3a. The microwave-integrated circuit (MIC) configured on a dielectric board consisted of a voltage-controlled oscillator (VCO) and multipliers in the transmission section, and a low-noise microwave/millimeter-wave amplifier and a detector in the receiver section. The output frequency of the VCO was 8–12 GHz and was multiplied up to 32–48 GHz using two multipliers. The microwave output suppressed to less than 1 mW was fed to a transmitting antenna. The sweep time of the incident frequency was 100 μsec, and the sweep cycle was 1 msec, which resulted in a duty cycle of 1/10. The reflected wave picked up by an identical antenna was fed to a

low-noise amplifier and a detector to provide the envelope of reflected power. Substrate (tapered-slot) antennas were used as a transmitter and a receiver, as shown in Figure 3b. The angle between the transmitting and the receiving antennas was 30 degrees (the incident angle was 15 degrees). Compared to the case of 0 degrees, this causes a 3–4% change in the peak frequency interval due to the Fabry–Perot interferometry effect. This does not affect the results. The distance between the antennas and the tiles was set to 5mm. Since the directivity of the antennas was poor, the reflected waves became very small at larger distances.

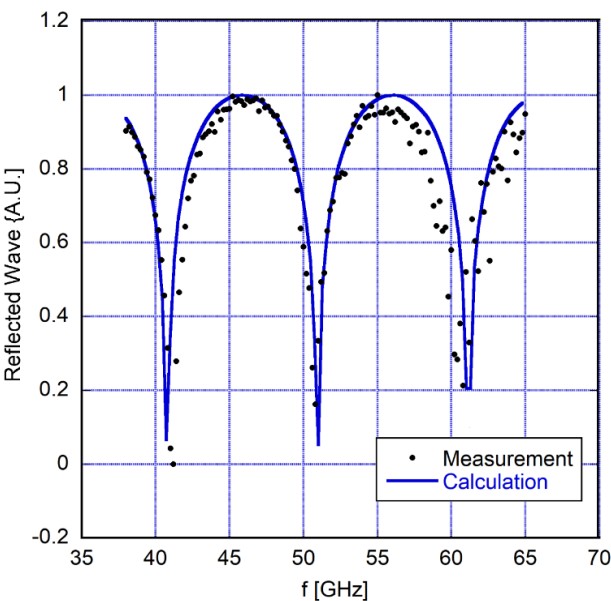

**Figure 2.** Measurement of reflected wave vs. frequency and numerical calculation of Fabry—Perot interferometry.

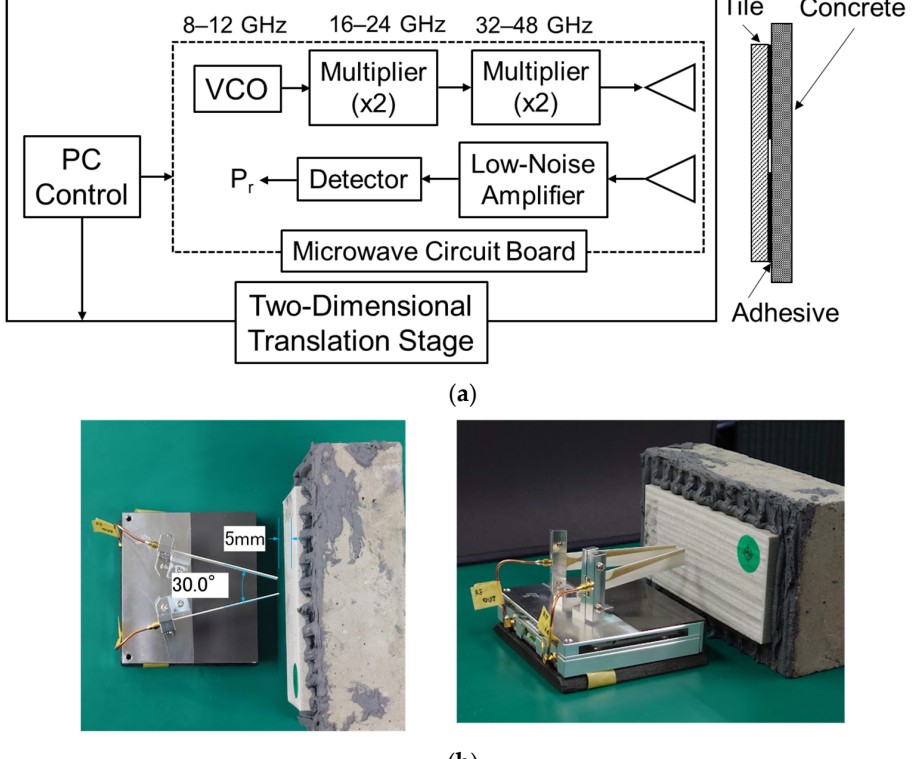

**Figure 3.** (**a**) Schematic of the prototype system; (**b**) picture of the transmitting and receiving antennas.

A PC control unit was responsible for identifying the measurement positions, transmitting the incident wave, detecting the reflected waves and quantitatively evaluating the adhesive state between a tile and the concrete base. The total system was installed on an X-Y stage. The signal for the frequency sweep emitted from the DA output of the control unit was fed to the microwave circuit, as well as the X-Y stage. The entire system was housed in a shield box in order to satisfy the weak power condition for leakage power from the system to be applied as a consumer appliance under the Administration of Radio, and the Ministry of Internal Affairs and Communications (MIC), Japan [14].

In the test experiment, the system consisting of an assembled MIC board and antennas, together with an automatic X-Y stage, was placed close to a tiled surface and scanned in a two-dimensional space. The scanning was performed every 1 mm in the *X-Y* axis using the meander scan style. Figure 4 shows the scanning process of the tiled surface. First, the tiled surface is scanned along the *X* axis from the initial position, then along the *Y* axis by 1 mm, followed by the negative *X* direction movement, and so on.

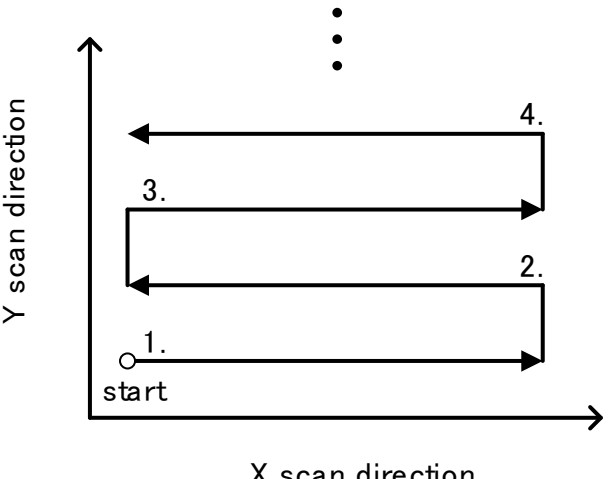

**Figure 4.** The trace of the two-dimensional scanning sequence.

*3.2. Signal Processing*

The overview flowchart of the signal processing algorithm is shown in Figure 5. The procedures were as follows: (i) Fixing the measurement positions, including the initial position. (ii) Performing reflectometric measurement; the received signal usually fluctuates due to the effect of multiple reflections between circuit elements (voltage standing wave ratio: VSWR > 1). This disturbs the evaluation of the signals. We measured a reference signal for a normal sample beforehand and input this into a processing algorithm in order to cancel those fluctuation components. This is rather similar to the calibration work of the VNA. (iii) Analysis of the reflectometer signal; the difference between the maximum and minimum values of the signal for the desired analysis frequency interval is measured, and the adhesive state is quantitively evaluated. The values are classified according to the "non-adhesive indication value" collected beforehand, with or without adhesive. (iv) Performing the above-mentioned processing for every measurement position. (v) After the measurement of the target area is completed, the adhesive area to the measured area, that is, the adhesion rate, is calculated.

**4. Experimental Results and Discussion**

Various types of simulation sample subjects (blocks with tiles attached to the surface) were fabricated for the practical test measurement by using the above prototype system. Specifically, the tiles were attached to the concrete surface using an organic adhesive which is difficult to detect by a hammering inspection. A standard sample is shown in Figure 6a. The tiles were glued on top of the block with adhesive and sandwiched between a 19 mm-wide cushioning material (polyethylene sheet). The thickness of the polyethylene sheet

was approximately 1 mm and played the role of an air layer. Another good example is shown in Figure 6b. It has a hollow space inside the block. As described, it is quite difficult to distinguish the air layer behind the tile from the hollow space of the concrete in these types of blocks using hammering methods, including a hammering robot.

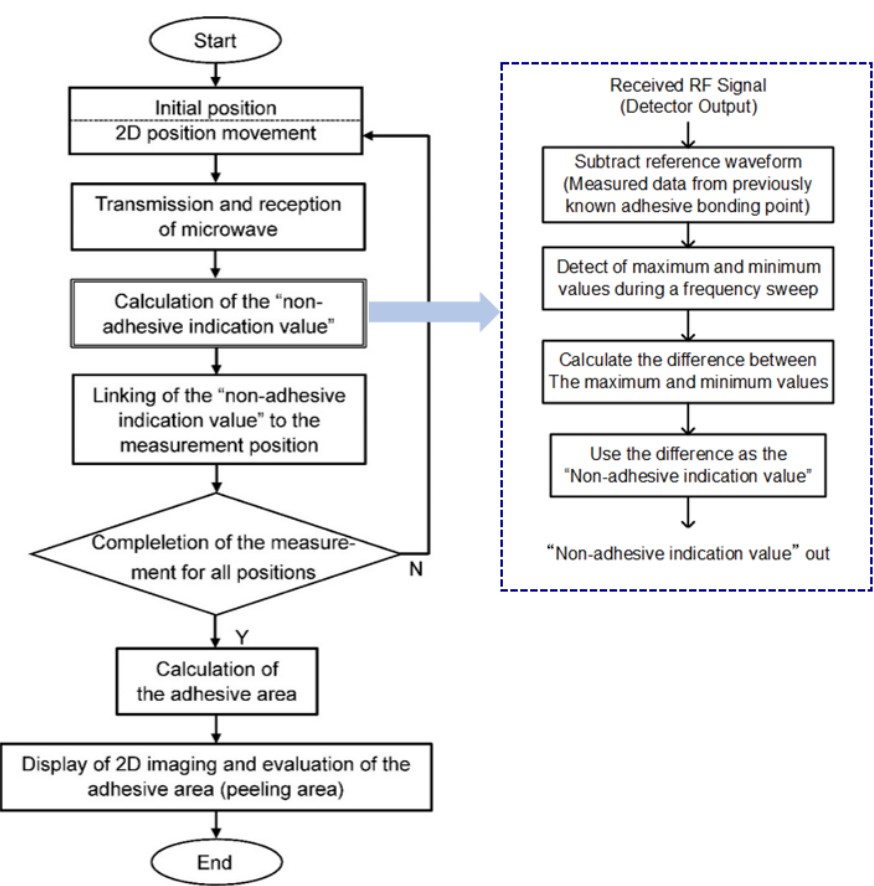

**Figure 5.** The overview flowchart of the signal processing algorithm.

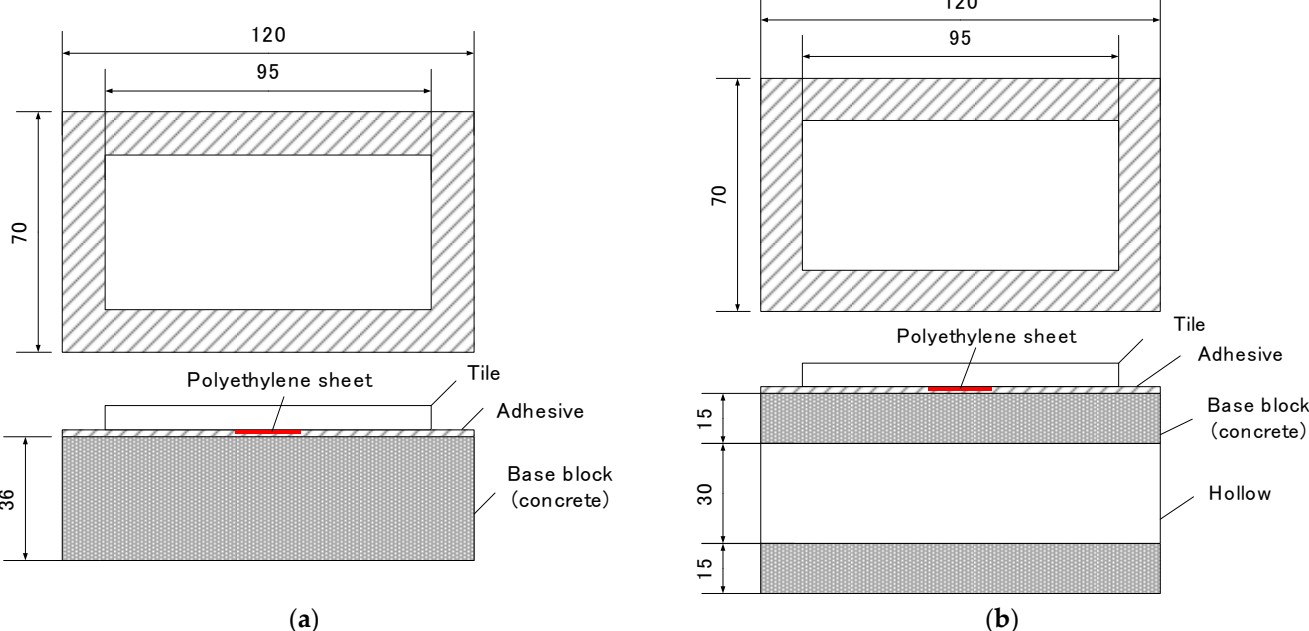

**Figure 6.** Test examples of the block: (**a**) normal type; (**b**) hollow type.

Figure 7a,b show the measurement results for the two samples (Figure 6a,b, respectively). For clarity, the area of a single tile is surrounded by a black frame. A threshold for the evaluation of the non-adhesive state was set from the difference between the maximum and minimum of the reflected wave signal. The yellow color area in the figure was determined to be the "non-adhesive state" by the microwave measurement.

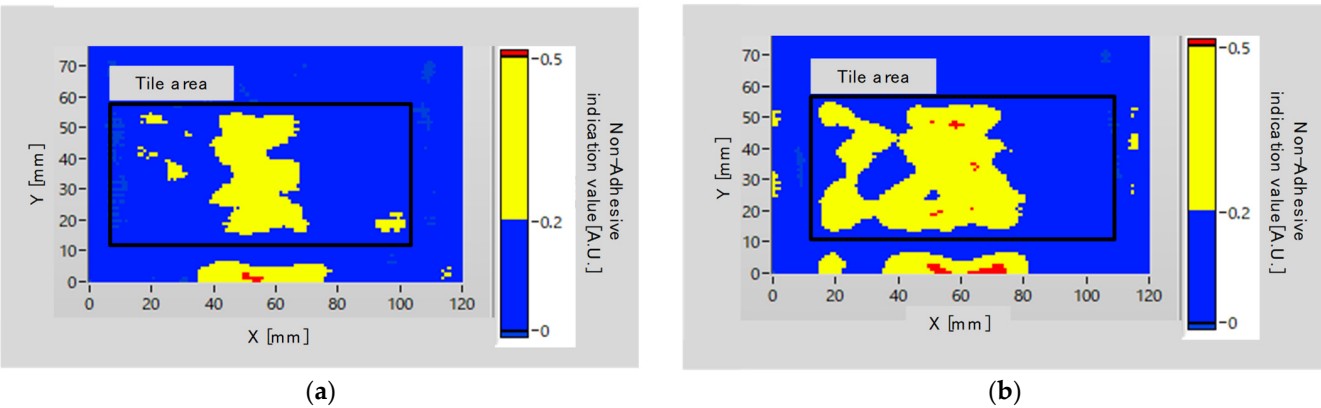

**Figure 7.** Microwave measurement: (**a**) normal base; (**b**) hollow-type base.

In order to verify the results of the microwave measurement, the conventional method was utilized; that is, the tiles were peeled off, and the adhesive surfaces were visually checked and classified into the following three categories for each 5 mm square.

a. Non-adhesive state: red dot.
b. Half-adhesive state: yellow dot.
c. Adhesive state: green dot.

The photos of the peeled surface are shown in Figure 8a,b. The results of the visual inspection are shown in Figure 8c,d. A cushioning material was inserted as a temporary air layer, but there were other areas where the air layer remained without being sufficiently bonded.

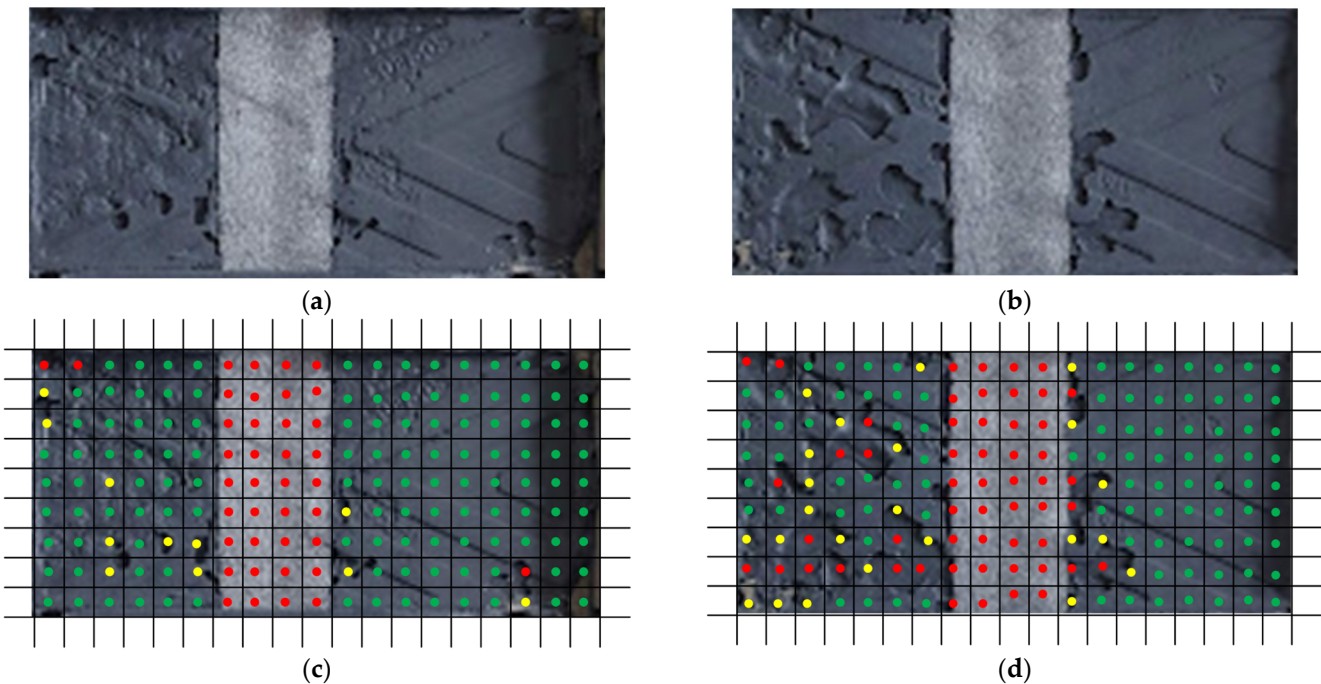

**Figure 8.** Visual results of bonding surface: (**a**) normal base; (**b**) hollow-type base; (**c**) normal base (visual evaluation overlaid); (**d**) hollow-type base (visual evaluation overlaid).

The adhesion area ratio, determined by the microwave measurement, was compared with the visual result, as shown in Table 1. The differences in the adhesion area ratio were 4.1 percent points (pp) for the normal sample and 2.0 pp for the hollow-type sample. These results are satisfactory. The present system can contribute to practical applications.

**Table 1.** Comparison of the adhesion rate between the fabricated system and visual inspection.

| Sample Type | Adhesion Rate Evaluated from Microwave Measurement | Adhesion Rate Evaluated from Visual Inspection | Difference between Microwave Measurement and Visual Inspection |
|---|---|---|---|
| Normal | 56.9% | 61.0% | 4.1 pp |
| Hollow | 77.0% | 75.0% | 2.0 pp |

## 5. Conclusions

In conclusion, a reflectometric diagnostic system was developed to measure the inside of a building wall, that is, to inspect tile materials attached to the building wall. The measurement principle utilized the phase interference effect of the reflected wave due to the multiple reflections between the two layers (Fabry–Perot interferometry effect). The results show the imaging inside the surface related to the non-adhesive condition between the tile and concrete wall, and the quantitative evaluation of the adhesion ratio area within an accuracy of 4.1% of the actual one. In addition, it is posited that the evaluation of an adhesive state with a hollow concrete base (extruded cement panel) and tiles with organic glue using conventional hammering inspection methods is difficult, including a hammering robot. However, there are no difficulties with the microwave/millimeter-wave reflectometric method, which indicates the superiority of the present method.

The development of a system such as the one proposed here is becoming important, since many constructions including buildings are aging. Since the present system has several advantages, it is expected to be applied to actual measurements in the field.

## 6. Patents

Yoshikazu Hirata, Atsushi Mase, Shota Osaki and Hiroshi Nakashima, The Outer Wall Diagnosis System, Japan, patent 6683964(2020).

**Author Contributions:** Conceptualization, A.M. and Y.H.; methodology, S.O., A.M. and M.I.; software, S.O.; writing—original draft preparation, S.O.; writing—review and editing, A.M.; supervision, Y.H. and M.I. All authors have read and agreed to the published version of the manuscript.

**Funding:** This research received no external funding.

**Institutional Review Board Statement:** Not applicable.

**Informed Consent Statement:** Not applicable.

**Data Availability Statement:** Data sharing is not applicable to this article.

**Acknowledgments:** The authors would like to thank to N. Machida (Manager, Production Control Section and Technology Development Office, Daisue Construction Co. Ltd., Tokyo 1368517, Japan) for his support and valuable discussion, A. Kusumi (Dedicated General Manager, Technical Division, Cemedine Co. Ltd., Tokyo 1418620, Japan) for his fruitful discussion and H. Tomonou (Manager, Technical Group, Kyushu Keisokki Co., Ltd., Fukuoka 8120015, Japan) for his beneficial discussion.

**Conflicts of Interest:** The authors declare no conflict of interest.

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
