# Peer review of "Imaging Diagnostics of Inside of a Building Wall Using Millimeter-Wave Reflectometer"

_applsci, doi:10.3390/app12062879_

Round 1

Reviewer 1 Report

Thanks to the authors for their interesting work and for submitting this manuscript, which addresses the use of microwaves for non-destructive inspection of building materials.

There are some comments and questions to the authors as follow:

  1. Generally, the paper has the structure of an industrial report rather than a scientific research paper. The theoretical part is weak and the overall structure is very much simplified.
  2. The introduction section is very general and does not include a wider view and proper literature review to address the cons and pros of microwave NDT and its applications as well as other NDT methods applied in building inspection. Moreover, the literature review is weak and needs to be improved considering some pioneering works.
  3. Section 2: theoretical explanations are not well explained and the words and expressions in some cases are wrong, e.g.:
  • Eq. 1 is the reflectance function and not the reflected electromagnetic wave, moreover, what is R2'?
  • Line 76: what is the significance of λ(ω)? 
  • As defined in Line 78 and Fig. 1, N is the refraction index, however, the authors didn't notice or understand that in Eq. 3. n is N and can not be n=1,2,3,... . 

4. In my opinion there's a difference between the concept described by the authors as the main concept of inspection with what happens in reality. The real case is a multilayer structure with various ε and therefore various n. Also, the difference in the ε of different martial layers shall be considered (tile, paste and concrete). There's no explanation about this in the manuscript. 

5. Authors have considered a fixed thickness for the tile (which fits into the frequency/wavelength range of the system). Since a microwave system (especially in reflectometric setup) is very sensitive to such changes, how this can be eliminated in results or how do you compensate for that?

6. In verification you have used an open-ended waveguide and a substrate one (not really clear) while in the real system you have used another type. Moreover, there's no information about the antenna and whether you are working in the near- or far-field and how these two configurations may affect the result.

7. Does Fig. 2 represent the reflectance? if so, the y-axis label shall be changed. Moreover, the system works in the frequency range of 32-48 GHz while the graph is from 35 to 65 GHz.

8. Fig. 3(b) is not clear. It has to be replaced with a high-quality image. Also as mentioned before, you haven't mentioned in what distance from the sample you did the scans and what was the incident angle, etc. You need to provide a detailed image and schematic.

9. Lines 132-135: is not clear to me it was a raster or meander scan, therefore, not clear the scan resolution was 1 mm or 2 mm?

10. Section 3.2 and Fig. 5 need more information on your thresholding method and how you handle the signals. Also, in Fig. 5 what is the 'Floating indicator out'?

11. Is that a proper choice to use a polyethene sheet to represent the air gap? please explain why?

12. What is the significance of using a hollow sample? The reason you mentioned (hammering inspection doesn't work well), is not in my opinion adding any challenge to the microwave inspection but it even improved the results. Can you explain why in your opinion the results are better?

13. Verification method represented in Fig. 8 is not making sense in my opinion. First, it more looks like a thermogram rather than an image taken by a photo camera. Second, after removing the tile, for sure some parts of the adhesive might be removed as well that may affect the visual inspection results. 

14. I was not able to find the Patent mentioned in Sec. 6 on any Japanese official website. Please cross-check or delete.

15. A fundamental question to authors: The microwave is having limitations, is a bit difficult to set up and understand by a normal operator, has radiation problems, it can be costly and so on, while for example methods like dry contact ultrasound or other ultrasound methods without the need for having couplant might be easily applied with high sensitivity to the airgaps. UT-based systems are easier to operate, are standardized and are more user-friendly than microwaves. Why hasn't been used or even compared with this technique?

Considering the abovementioned comments, I would like to ask the authors for a major revision of their manuscript.

Reviewer 2 Report

The manuscript presents a millimeter wave based reflectometer for detecting adhesion problems behind tiles during construction. The paper is well written and the contents are presented in a straight forward manner. Following are my questions: 1) What will be dielectric properties of the tile used in the experiments - in the frequency range of interest? 2) Will variation in the moisture content behind the tile affect the results and if it does how much? 3) How much does the variation in standoff distance or angle of incidence affect the results? I think these questions will come up during the practical application of the method and it will useful to address them.

Round 2

Reviewer 1 Report

Thanks to the authors for their detailed reply to all questions and comments. 

Although I am not very much convinced with the replies provided for comments #13 and #15, I accept the manuscript in its current form for publication.